# CD73/Adenosine Pathway Involvement in the Interaction of Non-Small Cell Lung Cancer Stem Cells and Bone Cells in the Pre-Metastatic Niche

**DOI:** 10.3390/ijms23095126

**Published:** 2022-05-04

**Authors:** Giulia Bertolini, Mara Compagno, Dimas Carolina Belisario, Cristiano Bracci, Tullio Genova, Federico Mussano, Massimo Vitale, Alberto Horenstein, Fabio Malavasi, Riccardo Ferracini, Ilaria Roato

**Affiliations:** 1Tumor Genomics Unit, Fondazione IRCCS Istituto Nazionale dei Tumori, 20133 Milan, Italy; giulia.bertolini@istitutotumori.mi.it; 2Center for Research and Medical Studies (CeRMS), A.O.U. Città della Salute e della Scienza di Torino, 10126 Turin, Italy; mara.compagno@gmail.com (M.C.); caro.belisario@gmail.com (D.C.B.); 3Laboratory of Immunogenetics, Department of Medical Sciences, University of Turin, 10126 Turin, Italy; cristiano.bracci@edu.unito.it (C.B.); horenstein.al@gmail.com (A.H.); 4Department of Life Sciences and Systems Biology, University of Turin, 10123 Turin, Italy; tullio.genova@unito.it; 5C.I.R Dental School, Department of Surgical Sciences, Bone and Dental Bioengineering Laboratory, University of Turin, 10126 Turin, Italy; federico.mussano@unito.it (F.M.); ferracini@edu.unige.it (R.F.); 6IRCCS Ospedale Policlinico San Martino Genova, UOC Immunologia, 16132 Genoa, Italy; massimo.vitale@hsanmartino.it; 7Department of Medical Sciences, University of Turin, 10124 Turin, Italy; fabio.malavasi@unito.it; 8Department of Surgical Sciences (DISC), University of Genoa, 16132 Genoa, Italy

**Keywords:** adenosine, non-small-cell lung cancer, cancer stem cells, bone metastases, osteoclast

## Abstract

Adenosinergic signaling is an important regulator of tissue homeostasis and extracellular accumulation of adenosine (Ado) and is associated with different pathologies, such as cancer. In non-small-cell lung cancer (NSCLC), a subset of CD133/CXCR4+ cancer stem cell (CSCs) has been demonstrated to initiate bone metastases. Here we investigated how NSCLC CSCs interact with osteoclasts (OCs) and osteoblasts (OBs) by modulating Ado production and OC activity. We proved that CSC-spheres, generated in vitro from NSCLC cell lines, express CD38, PC-1, and CD73, enzymes of the non-canonical adenosinergic pathway, produce high level of Ado, and down-regulate A1R and A3R inhibitory receptors, while expressing A2AR and A2BR. To address the Ado role and modulation of the in-bone pre-metastatic niche, we performed co-cultures of CSC-spheres with OCs and OBs cells. Firstly, we verified that active OCs do not activate non-canonical the adenosinergic pathway, conversely to OBs. OCs co-cultured with CSC-spheres increase Ado production that is related to the OC resorption activity and contributes to T-cell suppression. Finally, we proved the efficacy of anti-CD73 agents in blocking NSCLC cell migration. Overall, we assessed the importance of adenosinergic signaling in the interaction between CSCs and OCs at the pre-metastatic niche, with therapeutic implications related to Ado production.

## 1. Introduction

Adenosinergic signaling is an endogenous regulator of tissue homeostasis in both physiological and pathological conditions. In particular, accumulation of adenosine (Ado) in the extracellular compartment is associated with a variety of pathophysiological events in cancer [1,2,3], such as direct impairment of the anti-tumor immune response against cancer, and enforcement of the local immunosuppressive milieu [4]. The main biological action of Ado is mediated by G-protein-coupled cell-surface P1 purinergic receptors (R), A1R, A3R, A2AR, and A2BR. When stimulated, A1AR and A3AR inhibit adenylyl cyclase (AC), decreasing cAMP levels, indeed they are called inhibitory receptors. On the contrary, A2AR and A2BR activate AC, increasing cAMP levels and hence they are called stimulatory receptors [5]. CD73 (ecto-5′-nucleotidase, NT5E) is a cell surface ectoenzyme, and the main regulator of the production and magnitude of Ado in the microenvironment [6]. CD73 is the common link between the two main pathways (e.g., canonical and non-canonical) involved in Ado production [7]. The canonical pathway of Ado production originated from CD39 (ectonucleoside triphosphate diphosphohydrolase-1) activity and the conversion of ATP into ADP, then in AMP, the substrate of CD73. The non-canonical pathway, is where the catabolic activity is guided by CD38, which converts NAD+ to ADPR, which is then processed to AMP by PC-1/CD203a, and to Ado by CD73 [8,9]. An increased expression of CD73 has been observed in various cancers, such as non-small-cell lung cancer (NSCLC), gastric, renal, colorectal, and prostate cancers, and melanoma and often correlates with a poor prognosis [10,11,12,13,14,15,16]. CD73 overexpression in NSCLC has not only been associated with a poor prognosis in the primary tumor [12], but it is also correlated with EMT status [17] and an immunosuppression [18]. Indeed, myeloid-derived suppressor cells (MDSCs), expressing the ectonucleotidases CD39 and CD73, were able to inhibit T and NK cell activity in peripheral blood and tumor tissue of NSCLC patients [19,20]. More recently, it has been demonstrated that T-cell-mediated anti-tumor immunity can be rescued by suppressing the EGFR/CD73 axis in EGFR-mutated NSCLC [21].

We previously identified a subpopulation of NSCLC cancer stem cells (CSCs), characterized by the expression of CD133 and the chemokine receptor CXCR4, which was able to grow as spheres, and to initiate the metastatic process [22]. In a humanized mouse model, we provided evidence supporting the ability of CD133+/CXCR4+ CSCs-enriched spheres (from here-on named CSC spheres) to initiate bone metastases, that clinically occurred in approximately 30–40% of NSCLC patients [22]. This high incidence of bone localization by NSCLC was the stimulus for the study of CSC spheres, especially in regards of their potential cross-talk with bone cells, that are components of the bone pre-metastatic niche. More recently we showed that NSCLC CSC spheres do express high levels of CD73, which leads to the production of immunosuppressive Ado and the release of IL-10, contributing to the immunosuppression of the tumor microenvironment [23]. In the present study, we investigated how NSCLC metastatic CSCs spheres (CD133/CXCR4/CD73+ cells) interact with osteoclasts (OCs) and osteoblasts (OBs) by modulating Ado release and OC activity.

## 2. Results

### 2.1. The Non-Canonical Adenosinergic Pathway Is Responsible for Ado Release by NSCLC CSC Spheres

The expression of CD73 was increased in the subset of CSC spheres (Figure 1A) compared to the adherent counterpart (Figure 1B). In particular, the augment was significant for H3122 and A549 spheres (*p* < 0.05 (Figure 1C,D). The study of P1 purinergic receptor expression revealed that A1AR and A3AR were down-regulated in CSC spheres, while A2AR was not expressed by A549, but was, like A2BR, similarly expressed by adherent cells and CSC spheres of H2228 and H3122 (Figure 1D). These data suggest that NSCLC CSC spheres may decrease AC (thus inhibiting cAMP production) down-modulating A1AR and A3AR, to regulate Ado activity.

The expression analysis of ectoenzymes involved in the adenosinergic signaling revealed that the canonical pathway was not active, since CD39 was not expressed by NSCLC cells, either when grown in adherent condition or as spheres (data not shown). PC-1 expression was significantly up-regulated in spheres obtained from all three cell lines, *p* < 0.05. CD38 showed an increasing trend, even though it was only significant for H2228 spheres, *p* < 0.01. CD26 was expressed and did not show significant variations for H3122 or H2228, while it significantly increased in A549 spheres, *p* < 0.01 (Figure 1E). The observation may indicate bona fide that the non-canonical pathway was up-regulated in the CSC spheres. To confirm these observations, Ado released via the non-canonical pathway was assayed after adding AMP as substrate to adherent cells and CSC spheres. This assay proved that spheres derived from H3122 (64.6 ± 2.6 vs. 18.2 ± 0.02, mean area % of the peaks ± SD; *p* < 0.01) and A549 (44.2 ± 2 vs. 4.4 ± 0.6; *p* < 0.01) produced significantly higher level of Ado than their adherent counterpart (Figure 1F). The levels of Ado produced by H2228 spheres and adherent cells were quantitatively similar (27.3 ± 0.7 vs. 34.7 ± 5.7), (Figure 1E), concomitantly with the absence of significant differences for CD73 expression (Figure 1C).

### 2.2. CD73 and Purinergic P1 Receptors Are Differently Expressed by OCs and OBs

Looking at the expression of CD73 by TRAP+ OCs (Figure 2A) and ALP+ OBs (Figure 2B), a high level of protein was detected in OBs, while it was not present in OCs (Figure 2C). Between the purinergic AC inhibitory receptors, A3R was markedly expressed only by OCs, while A1R was slightly expressed by both OCs and OBs (Figure 2C). A2AR was expressed at similar level by both OCs and OBs, while A2BR was slightly expressed only by OBs (Figure 2C). OCs were able to produce a small amount of Ado (Figure 2C) and since they did not express CD73, Ado production is likely dependent on TRACP, an ectonucleotidase active in acidic environments and expressed by OCs. Conversely, OBs were able to produce Ado, likely mediated by the high expression of CD73 (Figure 2C), confirming the relevance of CD73/Ado for OB differentiation.

### 2.3. Ado Production Is Sustained by NSCLC CSC Spheres and OBs in Co-Cultures with OCs

To investigate the role of adenosinergic signaling in the context of bone metastatic niche formation, we assessed the modulation of Ado produced by OBs and OCs after the co-culture with CSC spheres (Figure 3A,B, respectively). Production of Ado by OBs was not significantly modified by the presence of CSC spheres (OBs 32.2 ± 1.8 vs. OBs + H2228s 29 ± 0.1; OBs + H3122s 25.2 ± 3.5; OBs + A549s 29.2 ± 1.8, Figure 3D). Since OBs are components of the bone metastatic niche, this high level of Ado production might contribute to the immunosuppressive microenvironment present in the niche. Conversely, co-cultures of CSC spheres and OCs resulted in an increase of Ado levels compared to OCs alone (OCs 4.4 ± 1.4 vs. OCs + H2228s 11.2 ± 0.2, *p* < 0.05; OCs + H3122s 24.7 ± 9.1; OCs + A549s 30.8 ± 2, *p* < 0.01, Figure 3C). As both OCs and OBs are present in the bone metastatic niche, OCs were co-cultured with OBs and CSC spheres all together. In addition, in the case of triple co-cultures, OCs showed significantly increased Ado production compared to OCs alone (OCs 4.4 ± 1.4 vs. H2228s + OCs + OBs 34.8 ± 5.4, *p* < 0.01; H3122 + OCs + OBs 16.3 ± 2.2; A549s + OCs + OBs 30.9 ± 1.8, *p* < 0.05, Figure 3E). These findings may suggest that the presence of CSC spheres induces an increased production of Ado by OCs in co-cultures, likely stimulating their activity.

### 2.4. NSCLC CSC Spheres Affect OC Activity and PBMC Proliferation

To verify whether a cell microenvironment enriched in Ado can interfere with OC activity, we quantified the mineral bone matrix resorption by OCs alone or co-cultured with CSC spheres. In the presence of H3122 and A549 spheres we detected an increased area of resorption, which was significant for A549 spheres, while with H2228 spheres the resorption area was reduced. Since OCs are active in an acidic environment, the reduction of pH in the microenvironment may induce an increase of Ado, that in turn can promote OC resorbing activity. Indeed, H2228 spheres, which synthesize low level of Ado, inhibited OC activity, at variance from the H3122 and A549 which produced high Ado levels and stimulated OCs (Figure 4A). Moreover, we quantified the TRACP activity of OCs, which is an index of their resorption activity, showing a tendency to increase in the presence of CSC spheres (Figure 4B), which may suggest that CSC spheres exert a stimulatory effect on OC activity.

To investigate the immune-suppressive properties of NSCLC CSC spheres, potentially impacting the development of bone metastasis and OC activities, we studied the effects of agents on T-cell proliferation of supernatants derived from spheres alone or co-cultured with OCs and in the presence of CD73. T-cell proliferation was significantly reduced in the presence of conditioned medium derived by NSCLC CSC spheres alone or co-cultured with OCs compared to the control, *p* < 0.05, Figure 4C. Supernatants from NSCLC CSC spheres treated with both APCP (*p* < 0.01) and the anti-CD73 moAb (*p* < 0.05) resulted in inhibition of T-cell proliferation.

### 2.5. Anti-CD73 moAb Inhibit Migration of NSCLC Spheres

Considering that CSC spheres enriched in CD133+/CXCR4+ CSC migrate in response to SDF-1, the ligand of CXCR4 highly expressed by bone cells, we tested the capability of CD73 inhibitory agents to interfere with cell migration. Both APCP 50 µM, and the anti-CD73 moAb at 20 µg/mL, significantly inhibited the migration of H2228 and A549 spheres, while the inhibition of H3122 sphere migration was not significant, Figure 5. This result suggests that blocking CD73 could be a way to prevent the SDF-1-mediated attraction of the CSC subset to the bone.

## 3. Discussion

High levels of CD73 expression have been reported in major subtypes of NSCLC [18]. Herein, we observed an increased CD73 expression in the subset of NSCLC stem cells, which grows as spheres and produce higher quantity of Ado compared to the adherent counterpart, confirming our previously published data [23]. Ado generated by hydrolysis of AMP can be accumulated in the extracellular environment and bind P1 purinergic receptors, or it can be subjected to a rapid conversion into inosine by ADA. A further alternative is its internalization by nucleoside transporters [24,25]. Ado effects depend on the type of targeted tissue and the number of expressed purinergic receptors. For instance, a higher number of receptors and higher affinity (A1 and A2A are of high affinity and A3 and A2B are of lower affinity for Ado) increase the activity or response to Ado [26]. Our NSCLC CSC-enriched spheres down-regulated A1R and A3R cAMP inhibitory receptors compared to the adherent counterpart, thus regulating Ado activity, concomitantly with the expression of A2AR and A2BR stimulatory receptors. Indeed, literature data report how A2AR and A2BR, as well as CD73 enzyme, are often overexpressed in primary tumors and metastasis [27,28]. Furthermore, the co-inhibition of CD73 and P1 purinergic receptors has been tested and found to be effective in the control of metastatization [29,30].

Looking at the expression of adenosinergic molecules involved in both the canonical and non-canonical pathways of Ado production, we did not detect CD39 on NSCLC cells, either when grown in adherent condition or as spheres, suggesting that the canonical pathway for Ado production was not present in these cells. Conversely, CD38 and PC-1 were highly expressed and up-regulated in the CSC spheres compared to adherent cells. CD26, an ectoenzyme involved in the transformation of Ado into inosine, was expressed by both adherent cells and CSC spheres, not showing significant variations in H3122 or H2228, while it was significantly increased in A549 spheres. Altogether, these results show that the Ado non-canonical pathway is activated in NSCLC CSC

Ado is an important regulator of bone homeostasis, indeed, patients with ADA deficiency have skeletal abnormalities [31], while ADA-deficient mice show a decreased trabecular bone [32]. CD73-KO mice show a decreased bone mineral content due to a reduction of Ado production, with consequent inhibition of OB differentiation [33]. Human OB precursors produce extracellular Ado, which modulates the secretion of interleukin 6 (IL-6) and osteoprotegerin, contributing to the regulation of bone resorption and formation [34]. The differentiation state of the cell influences the release of ATP, a primary substrate of canonical adenosinergic enzymes in bone. Indeed, mature OBs are reported to release higher amount of ATP than undifferentiated cells [35,36]. In our cultures, we showed that active OCs (generated on mineral matrix plates, mimicking bone) did not express CD73 and were characterized by the production of small amounts of Ado. This Ado production was likely dependent on TRACP, an ectonucleotidase expressed by OCs, which operates in acidic conditions, where CD73 is inactive [37]. Conversely, OBs express high levels of CD73 along with a high amount of Ado production. In line with literature data, our findings support the essential role of CD73 in OB activity and differentiation [33]. Looking at the expression of the AC inhibitory purinergic receptors, OBs did express only A1R, while OCs expressed both A1R and A3R, confirming previous data [38]. In rat models, stimulation of A3R reduced the number of OCs [39] and inhibited bone metastases induced by breast cancer cells [40]. Differently from our data, Evans et al. reported that OBs do express both A1R and A3R [34]. A possible explanation for the discrepancy is that they used a cell line derived from human bone marrow stromal cells, expressing osteoblastic markers, while we generated OBs starting by ASC52telo, cultured in osteogenic medium. Thus, the different adopted model might explain the discordant observation. Further Gharibi et al., while investigating the role of P1 receptors in mesenchymal stem cell (MSC) differentiation, reported that A1R is expressed during adipogenesis, while its expression is low in osteogenesis [41]. Besides, Kara et al. demonstrated that the activation of A1R modulates osteoclastogenesis [42]. However, no significant changes were observed in bone formation, suggesting that A1R is not fundamental for OB activity [43].

A2AR was expressed by both OCs and OBs, while A2BR was expressed only by OBs. The reports on the effects of A2AR activation of in OCs are contrasting. Indeed, both an inhibitory and stimulatory effects on OC formation [44] and differentiation have been reported [45,46]. The role of A2AR in OBs is ambiguous since some authors reported an increase in osteoblastogenesis after its stimulation [45] and enhanced bone regeneration [47]. In contrast, other groups did not observe effects on differentiation and mineralization due to A2AR modulation [48]. A2BR is up-regulated in the early phase of osteoblastogenesis [41,49]. Moreover, it has been demonstrated that Ado promotes osteoblastogenesis primarily through A2BR, mediated by an up-regulation of the expression of RUNX2 [50,51,52]. Our OCs were active in resorbing matrix and did not express A2BR, which inhibits OC differentiation and reduces OC activity [38,45].

A relevant point is represented by the potential effect of CSC spheres on modulating the levels of Ado in the presence of bone cells, such as OCs and OBs. Consequently, we performed co-cultures of OCs and OBs with NSCLC CSC spheres, reporting contrasting results in terms of Ado production. OCs, which by themselves produced small amount of Ado, since they did not express CD73, were induced to significantly increase Ado production by NSCLC CSC spheres. By contrast, co-cultures of OBs and CSC spheres did not result in significant variations of Ado production. To further simulate the bone metastatic niche, we co-cultured OCs, OBs, and NSCLC spheres all together, then we dosed checked whether Ado production by OCs was modified after the co-culture period, observing significant Ado increase compared to OCs alone. TRACP activity can be responsible for Ado production, and it is correlated with OC activity, thus we measured this enzyme in cultures of OCs alone or OCs with NSCLC CSC spheres, showing an increasing trend of production in the presence of CSC spheres. We evaluated whether the presence of NSCLC CSC spheres may interfere with OC activity in terms of mineral matrix resorption. We detected a significant increased resorption only in presence of A549 spheres, and a significant decrease with H2228 spheres, since the increase of Ado, that lowers pH in the microenvironment, can promote OC resorbing activity, that is activated in acidic environment. Indeed, co-culture of OCs with H3122 and A549, which produce high Ado levels stimulated OCs, conversely to H2228 spheres, that generated low levels of Ado, inhibited OC activity. The subset of CD133/CXCR4/CD73+ NSCLC stem cells are endowed with immune-suppressive properties due to the release of cytokines and Ado able to activate immune cells which promote growth of cancer cells [23]. To mimic the immunosuppressive microenvironment occurring during bone metastasis formation, we tested whether supernatants of CSC spheres cultured alone or with OCs could influence PBMC proliferation. Both NSCLC CSC spheres and OCs, cultured alone or co-cultured, secreted supernatant factors able to inhibit PBMC proliferation. The treatment of NSCLC CSC spheres with CD73 inhibitors did not revert the inhibition of PBMC proliferation; on the contrary, it was maintained. This suggests that factors other than Ado are responsible for this inhibitory action on immune cells.

Other than its enzymatic activities, CD73 could act as a co-stimulatory signaling molecule, regulating cell adhesion properties, and influencing cell migration and invasion processes during the EMT process observed in different cancers [53,54,55]. Previous results from our group, showed that NSCLC stem cells, the CD133+/CXCR4+EpCAM subset, was triggered by EMT, an event that can be induced by stimuli from the tumor microenvironment, able to convert progenitor cells into cancer stem cells [22], endowed with invasiveness ability [56,57]. We investigated the ability of anti-CD73 agents to interfere with the migration of NSCLC cells. The results indicated that both APCP and the anti-CD73 moAb significantly inhibited migration of H2228 and A549 cells, in line with previous observations on breast cancer cells, characterized by inhibition of both tumor growth and metastasis [55,58,59].

In conclusion, our findings provide important hints on additional functional interactions between CSC spheres and OCs in the pre-metastatic niche, with important therapeutic implications related to Ado production. Indeed, the interaction of Ado with its receptors can lead to an immunosuppressed niche in the tumor microenvironment, which can be a target for new therapies [60]. Further studies will help to deeply investigate the immunologic consequences of this cross-talk, and the possible functional variability related to the heterogeneity of the NSCLC histotypes and the EMT status of CSC spheres. Our “human-in-mice” setting of bone metastasis will represent a reliable support to validate, in vivo, possible therapeutic agents targeting CSC-OC-immune-cell cross-talk. Thus, for example, in the murine model, it will be interesting to assess whether anti-CD73 agents can block the bone metastatic process induced by NSCLC spheres. To date, several CD73-blocking antibodies recently entered human cancer trials in monotherapy or in combination with checkpoint inhibitors, and conclusive results are awaited [61]. Indeed, even though immunotherapy has improved survival of patients with lung cancer [62], a high rate of resistance still limits its efficacy [63]. Among the different mechanisms regulating the resistance to the checkpoint inhibitors, the up-regulation of CD38 by tumor cells causes a functional impairment of CD8 T cells, favoring the tumor immune escape. Thus, the contemporary inhibition of immune checkpoints and Ado release improves the anti-tumor immune response [20] and should a novel strategy to improve treatment effectiveness of lung cancer patients.

## 4. Materials and Methods

### 4.1. NSCLC Cells and CSC Sphere Cultures

NSCLC cell lines (A549, H2228, and H3122) were purchased from ATCC and cultured in RPMI-1640 supplemented with 10% heat-inactivated bovine serum (FBS, all from Lonza, Basel, Switzerland).

To originate NSCLC CSC sphere cultures, cells were plated in Ultra-Low Attachment plates (Corning, NY, USA) at low density of 10^4^ cells/mL, to avoid cell aggregation and to assure clonal origin of generated spheres, in a stem cell medium (SCM): serum-free medium DMEM/F12 (Lonza, Basel, Switzerland), supplemented with B27 (Gibco, Billings, MT, USA), EGF 20 ng/mL, bFGF10 ng/mL (PeproTech, East Windsor, NJ, USA), and heparin 2 µg/mL, as previously described [64]. Sphere cultures were expanded for 15 days, then tested in the different experimental settings.

### 4.2. Flow Cytometry Analysis

The expression of NSCLC cell markers and molecules involved in the adenosinergic pathways were evaluated by flow cytometry. Adherent cells were detached through trypsin and spheres were dissociated using accutase solution (Sigma-Aldrich, St. Louis, MO, USA), then the single-cell solution was washed in staining buffer (PBS 1X + 0.5% BSA + 2mM EDTA) and incubated for 20 min at RT with the following anti-human antibodies: PE-CD133/1 (Miltenyi Biotech, Auburn, CA, USA), APC-CXCR4 (Miltenyi Biotech, Auburn, CA, USA), FITC-CD73, and PE-PC1 (kindly provided by the Malavasi Lab.), CD38 (Miltenyi Biotech, Auburn, CA, USA), CD26 (eBioscience, San Diego, CA, USA), and CD39 (BD Biosciences, Franklin Lakes, NJ, USA). Isotypic controls and unstained samples were used to set the negative control. Data were acquired using a MACsQuant 10 cytometer and analyzed through MACsQuantify software (Miltenyi Biotech, Auburn, CA, USA).

### 4.3. Western-Blot Analysis

Whole-cell extracts were obtained from adherent cells and CSC spheres using GST-FISH buffer (10 mM MgCl2, 150 mM NaCl, 1% NP-40, 2% glycerol, 1 mM EDTA, and 25 mM HEPES pH 7.5) supplemented with protease inhibitors (Roche), 1 mM phenylmethanesulfonylfluoride (PMSF), 10 mM NaF, and 1 mM Na3VO4. Extracts were cleared by centrifugation at 12,000 RPM for 15 min. The supernatants were collected and assayed for protein concentration using the Bio-Rad protein assay method. Twenty μg of proteins was loaded on 12% Mini-PROTEIN TGX gels (Biorad, Hercules, CA, USA), transferred on nitrocellulose membrane (GE Healthcare, MA, USA), and blocked with 5% skim milk (Biorad, Hercules, CA, USA). Primary antibodies for immunoblotting included monoclonal anti-rabbit NT5E/CD73 (D7F9A clone, 75KD, Cell Signaling Technology, Danvers, MA, USA), rabbit polyclonal anti-Adora1 receptor (A1R, 37KD), Adora2A and 2B receptors (A2AR, 44KD; A2BR, 36KD) from ElabScience (Houston, TX, USA), Adora3 receptor (A3R, 36KD) from ThermoFisher (Waltham, MA, USA), and anti-β-actin (Sigma-Aldrich, St. Louis, MO, USA). Membranes were developed with ECL solution (GE Healthcare, Westborough, MA, USA).

### 4.4. OB Generation

OBs were differentiated starting by an adipose-derived mesenchymal stem cell line, ASC52telo, hTERT immortalized, purchased from ATCC. These cells were expanded in Mesenchymal Stem Cell Basal Medium (ATCC PCS-500-030) added with Mesenchymal Stem Cell Growth Kit (ATCC PCS-500-040). To induce osteoblastogenesis, ASC52 telo were maintained in alpha-MEM supplemented with 10% FBS, 50 µg/mL ascorbic acid, 10^−8^ M dexamethasone, and 10 mM beta-glycerophosphate (Sigma-Aldrich, St. Louis, MO, USA). The in vitro differentiation of ASC52 into OBs was assessed through alkaline phosphatase staining, using the alkaline phosphatase kit by Sigma-Aldrich, according to the manufacturer’s instructions.

### 4.5. OC Generation

PBMCs derived from buffy coats were isolated after centrifugation over a density gradient by the Ficoll method and plated in 24-well Osteoassay plates (Corning, NY, USA) at 5 × 10^5^ cell/well, in Alpha-Minimal Essential Medium (α-MEM, supplied by Lonza), supplemented with 10% fetal bovine serum, benzylpenicillin (100 IU/mL), and streptomycin (100 mg/mL; Lonza, Basel, Switzerland), and maintained at 37 °C in a humidified atmosphere of 5% CO_2_.To induce the formation of osteoclasts (OCs), recombinant human M-CSF (25 ng/mL) and RANKL (30 ng/mL; PeproTech, East Windsor, NJ, USA) were added in culture. After 15 days, cells were stained for tartrate-resistant acid phosphatase (TRAP staining kit was supplied by Cosmo Bio Co., Ltd., Tokyo, Japan) and OCs were identified as TRAP-positive multinucleated cells containing three or more nuclei. Osteoassay plates provided a biomimetic surface, useful to measure OC activity, which was quantified through a semi-automated image analyzing system, and the dosage of TRAP cell-culture supernatants was quantify by an ELISA reader.

### 4.6. Co-Cultures of OCs, OBs, and CSC Spheres

Co-cultures of OCs with NSCLC CSC spheres were performed on Osteoassay plates, where OCs were allowed to differentiate and activate for 15 days, then 1. 5 × 10^4^ NSCLC CSC spheres were added in the wells, where OCs were grown. ASC52 were allowed to differentiate into OBs for 15 days onto a Transwell^®^ (BD Falcon, Franklin Lakes, NJ, USA) insert (1 × 10^4^ cells/insert), then the inserts were transferred in the wells of OCs and CSC spheres. All the co-cultures were maintained for 5 days, then the medium was collected to obtain conditioned medium of the different settings and Ado production was immediately assayed on OCs, deprived of CSC spheres and/or OBs.

### 4.7. Adenosine Quantification

Twenty-four hours before the Ado assay, NSCLC adherent cells were seeded on 24-well plates at a concentration of 5 × 10^5^/500 µL, while CSC spheres were transferred into 24-well plates in new medium, after having been cultured for 15 days (as previously described). Culture medium was removed from adherent cells by pipetting and from CSC spheres by centrifuging at low speed to pellet them down. Adherent cells and NSCLC CSC spheres were incubated with 100 µL STOP solution (constituted by erythro-9-(2-hydroxy-3-nonyl) adenine EHNA 100 µmol/L, dypiridamole DYP 10 µmol/L, 30 μg/mL levamisole, and 10 µmol/L deoxycoformycin DCF, an adenosine deaminase (ADA) inhibitor) by Sigma-Aldrich for 15 min at 37 °C, then treated with 100 µL AMP 100 µmol/L for 10 min at 37 °C on a tilting platform. After incubation, the cells were collected in a tube containing acetonitrile (ACN; 1:2; 4 °C) and centrifuged (13,000× *g* for 5 min at 4 °C). Then, tubes were transferred into a Speed Vac (Eppendorf) to remove the supernatant, reconstituted in HPLC-grade water, and assayed or stored at −80 °C. HPLC (Beckman) fitted with a reverse-phase column (Synergi 4U Polar-RP80A; 150 × 4.6 mm; Phenomenex) was utilized for the chromatographic analyses of the supernatant. Nucleotides and nucleosides were separated using a mobile-phase buffer (0.025 mol/L K2HPO4, 0.01 mol/L sodium citrate, and 0.01 mol/L citric acid, adjusted with phosphoric acid to a pH of 5.1 and 8% acetonitrile (ACN) for 13 min at a flow rate of 0.6 mL/min. Ultraviolet (UV) absorption was measured at 254 nm. Chromatography-grade standards used to calibrate the signals were dissolved in PBS 1X, pH 7.4 (Sigma-Aldrich), 0.2 µm-filtered, and injected in a volume of 15 µL. The retention times (Rt, in min) of standards were: AMP, 5.8; inosine (INO), 6.4; and adenosine (ADO), 10; using a Rt window of ±5%. Peak area was calculated using Gold software (Beckman, Brea, CA, USA). Quantitative measurements were inferred by comparing percentage area of each nucleotide and nucleoside analyzed, as previously described [65].

### 4.8. PBMC Proliferation Assay

PBMCs derived from buffy coats were plated in a 96-well plate at 2 × 10^5^ cells/well in RPMI, 10% FBS. To induce T-cell proliferation, PBMCs were stimulated with anti-CD3, OKT3 (7.5 μg/mL), and anti-CD28 (7.5 μg/mL). To test the effects of CSC spheres alone or co-cultured with OCs on T-cell proliferation, the stimulated PBMCs were cultured with 50% of their conditioned medium for 72 h. According to cell lines, negative control of the experiment was T cells cultured in RPMI 10% or 50% SCM. MTT assay was performed to assess T-cell proliferation, according to the manufacturer’s instructions (Sigma-Aldrich, St. Louis, MO, USA).

### 4.9. Migration Assay

For migration assays, 50 × 10^3^ cells/well were incubated with anti-CD73 monoclonal antibody (moAb) at 20 µg/mL or adenosine 5′-(α,β-methylene) diphosphate (APCP, at 50 µM, Sigma-Aldrich, MO, USA). The moAb anti-CD73, clone AD2, was generated and purified in-house through a two-steps HPLC chromatography in the Malavasi Laboratory. Cells were seeded in RPMI-1640 medium supplemented with 1% FBS onto 8 μm-pore Transwell^®^ cell-culture inserts (BD Falcon, Franklin Lakes, NJ, USA), in 24-well plates. The chemoattractant factor used was SDF-1 (50 ng/mL), put in RPMI in the lower chamber. After 48 h, cells on the top of the insert membranes were removed by scraping with a sterile cotton swab, while migrated cells on the lower side of the insert were fixed in methanol and mounted on slides using the VECTASHIELD mounting medium, containing DAPI. Five random fields were counted by fluorescence microscope visualization at 20× magnification for each insert. For every cell line, experiments were performed in triplicate.

### 4.10. Statistical Analyses

Statistical analyses were performed using GraphPad Prism version 6.0. Two-sided Student’s t-test was utilized to detect statistically significant difference between two groups. Statistical analysis among more than two groups was performed by one-way Anova with Tukey’s post-hoc test. Student’s *t*-test was used for analysis between two groups. Data are expressed as means ± standard deviation (SD), unless otherwise indicated. Statistical significance was defined as a *p*-value less than 0.05.

## Figures and Tables

**Figure 1 ijms-23-05126-f001:**
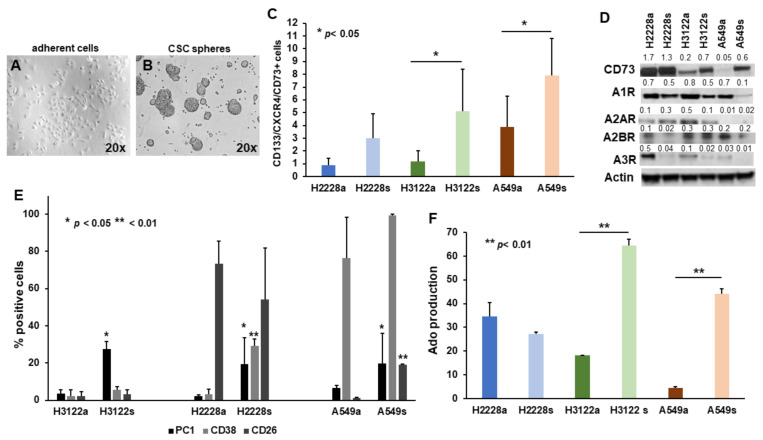
CD73 and Ado production by NSCLC cells. Images show NSCLC cell lines grown as adherent cells (**A**) and as as spheres, when cultured in the specific serum-free medium (**B**). (**C**) The subset of stem cells expressing CD133/CXCR4 was also highly positive for CD73, with a significantly increased expression for H3122s and A549s compared to the adherent counterparts, *p* < 0.05. (**D**) Western blot analysis confirmed the up-regulation of CD73 in NSCLC spheres compared to the adherent counterpart and showed the expression of the different Ado P1 purinergic receptors. Quantification of each band is reported (**E**). The graph reports the level of expression CD38, PC1, and CD26, which were basically up-regulated in NSCLC spheres compared to the adherent cells; data are the mean value ± SD of three analyses for each cell line. (**F**) Bars indicate the quantification of Ado production, as mean area % of the peaks ± SD, showing a significantly increased Ado production by H3122 and A549 spheres, *p* < 0.05. Student’s *t*-test was performed to compare adherent cells and spheres.

**Figure 2 ijms-23-05126-f002:**
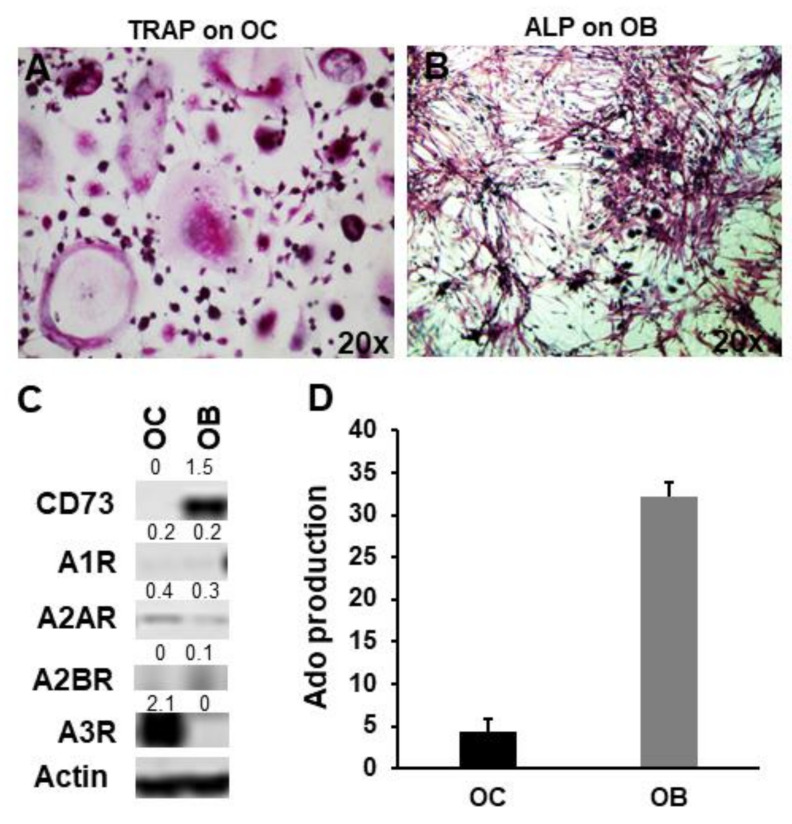
CD73 and Ado production by OBs and OCs. (**A**) OCs positive for TRAP staining and (**B**) OBs for alkaline phosphatase (magnification 20×). (**C**) Western blot analysis of CD73 and Ado P1 purinergic receptors is shown. Quantification of each bar is reported. (**D**) Bars indicate the quantification of Ado production, as mean area % of the peaks ± SD, by OCs and OBs.

**Figure 3 ijms-23-05126-f003:**
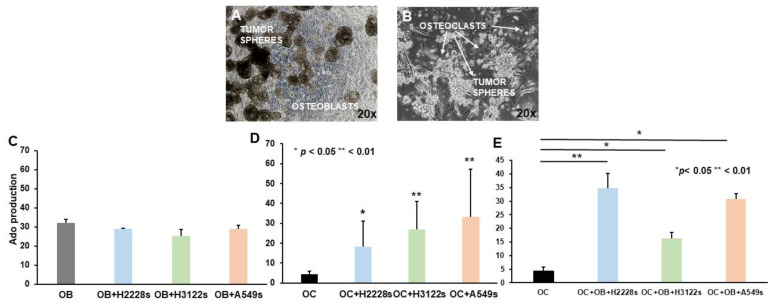
Ado production in co-cultures of NSCLC CSC spheres and bone cells. Representative images of NSCLC CSC spheres co-cultured with OBs (**A**) and OCs (**B**) are shown (magnification 20×). The graphs show the quantification of Ado production expressed as mean area % of the peaks ± SD, in the different culture settings (**C**–**E**). The differences in Ado production among the three types of CSC spheres, OBs, and OCs was analyzed by one-way Anova.

**Figure 4 ijms-23-05126-f004:**
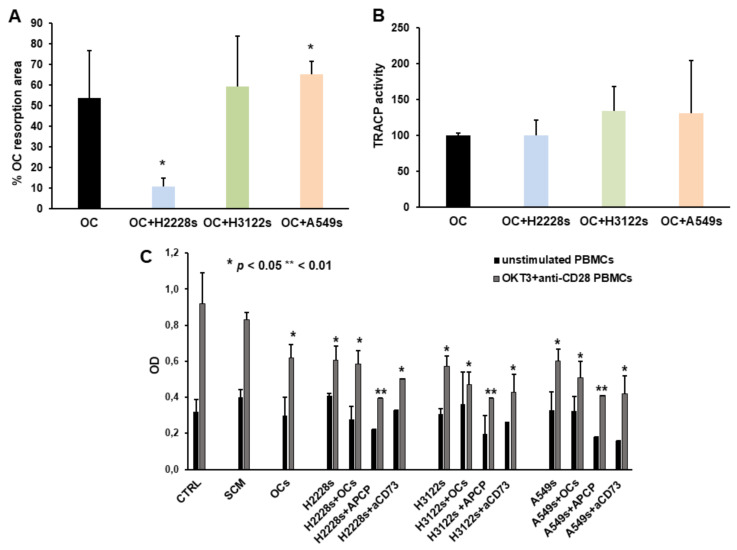
Effect of NSCLC CSC cells on OC activity and T-cell proliferation. (**A**) Bars indicate the activity of OCs, expressed as mean area of resorption ± SD. The presence of H3122 and, above all, of A549 NSCLC CSC spheres stimulates OCs, *p* < 0.05, while H2228s inhibit OC activity. (**B**) The release of TRAP by OCs was slightly increased in the presence of NSCLC spheres. (**C**) The assay measured the effect of the conditioned medium, derived from the different culture conditions, on T-cell proliferation, unstimulated and stimulated with anti-CD3 (OKT3) and anti-CD28 antibodies for 72 h. CTRL indicates T-cell proliferation in RPMI 1% FBS, SCM is the stem MTT cell medium. Data are represented as mean value ± SD of four independent experiments from each cell line. * indicates the significance compared to the CTRL calculated by one-way Anova analysis.

**Figure 5 ijms-23-05126-f005:**
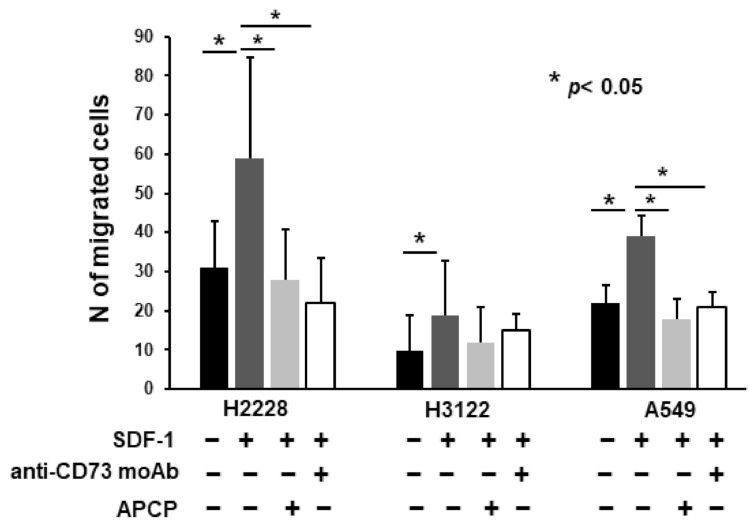
In vitro migration assay ofH2228, H3122 and A549 cell lines. Cells were treated with CD73 inhibitors, moAb anti-CD73 at 20 µg/mL and APCP at 50 µM, and chemoattracted by SDF-1 at 50 ng/mL. Data represent the mean number ± SD of migrated cells after treatment relative to the untreated control. Significance was calculated by one- way Anova comparing treated cells vs. positive control. Two experiments were performed for each cell line.

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
