# Peer review of "CD73/Adenosine Pathway Involvement in the Interaction of Non-Small Cell Lung Cancer Stem Cells and Bone Cells in the Pre-Metastatic Niche"

_ijms, 2022, doi:10.3390/ijms23095126_

Round 1
Reviewer 1 Report
The manuscript is well written and scientifically sound. I only have minor questions and concerns to be addressed by the authors, which go as follows:
Line 147: how were the cultures deprived of CSC spheres, after they grew in the same wells together with OCs?
Lines 204-205, Fig.1A: Add picture of adherent cells for comparison
Fig.2A: Same: I’m not sure the figure is illustrative if shows only positive staining, without some control that doesn’t stain positive.
Fig. 3 A and B: the red font on the pictures is not readable
Also the resolution of the figures should be improved, for example it is not always clear and needs to be zoomed whether it is H3122s or H3122a; same for all cell lines on many of the diagrams, specifically Fig.1 B and C and Fig. 3 all diagrams.
Chapter 3.1 shows that NSCLC CSC spheres produce Ado. Why the production of Ado by NSCLC CSC spheres was not measured in the presence of OCs and OBs in chapter 3.3? Based by data, there may be a possibility that they also produce more Ado in these conditions and contribute to the whole microenvironment?
Lines 258-259, also Fig. 4: The figure (green bar vs. black bar) does not illustrate an increase of OC activity by H3122 cells, as it is stated in the text.
Line 264: “The slight increase” does not look close to being statistically significant; I would suggest being more careful with the conclusions.
Fig.5: What was the control – completely untreated cells (as the legend states) or SDF-1+ cells? The bars that are showing significance do not differ from the untreated control that is black bar. Please explain.
Author Response
We thank the Editor and the Reviewers for handling and carefully revising our submission and for providing valuable suggestions that helped us to improve the manuscript.
In the revised version we have tried to address all points and concerns raised by the reviewers, by better elucidating and explaining technical issues and data interpretation.
Please find below a point-by-point reply to reviewers’ questions.
The manuscript is well written and scientifically sound. I only have minor questions and concerns to be addressed by the authors, which go as follows:
Line 147: how were the cultures deprived of CSC spheres, after they grew in the same wells together with OCs?
R: At the end of the co-culture experiments, media with CSCs growing in suspension were removed by pipette and then Ado was dosed. Thus, OCs were not further cultured after the co-culture period.
Lines 204-205, Fig.1A: Add picture of adherent cells for comparison.
R: We have added a representative picture of adherent cell line
Fig.2A: Same: I’m not sure the figure is illustrative if shows only positive staining, without some control that doesn’t stain positive.
R: Since TRAP and ALP are markers specifically expressed by entire population of OC and OB respectively, the observed positivity for TRAP and ALP is the confirmation that we obtained in vitro functional OCs from PMBC using Osteoassay plates, and OBs from mesenchymal stem cells, cultured in osteogenic medium. Since after differentiation OCs and OBs are respectively positive for TRAP and ALP, we cannot show any not stained control cells. We believe that this picture is just supportive of a well-standardized method for OC and OB generation that wouldn’t require negative controls.
Fig. 3 A and B: the red font on the pictures is not readable. Also, the resolution of the figures should be improved, for example it is not always clear and needs to be zoomed whether it is H3122s or H3122a; same for all cell lines on many of the diagrams, specifically Fig.1 B and C and Fig. 3 all diagrams.
R: We apologize for the poor resolution of figures and legends, we have modified and improved them
Chapter 3.1 shows that NSCLC CSC spheres produce Ado. Why the production of Ado by NSCLC CSC spheres was not measured in the presence of OCs and OBs in chapter 3.3? Based by data, there may be a possibility that they also produce more Ado in these conditions and contribute to the whole microenvironment?
R: According to chapter 3.3., the activity of CSC was measured in the presence of OCs and OBs (Fig. 3E) and indeed, they produce more Ado in these conditions and contribute to the whole microenvironment, as suggested by the Reviewer. (see data Fig.3E): OCs 4.4 ± 1.4; vs. H2228s + OCs + OBs 34.8 ± 5.4; vs. H3122 + OCs + OBs 16.3 ± 2.2; vs. A549s + OCs + OBs 30.9 ± 1.8).
Lines 258-259, also Fig. 4: The figure (green bar vs. black bar) does not illustrate an increase of OC activity by H3122 cells, as it is stated in the text.
R: We apologize for the mistake, we have now corrected the text
Line 264: “The slight increase” does not look close to being statistically significant; I would suggest being more careful with the conclusions.
R: We agree with the reviewer, and we corrected the sentence
Fig.5: What was the control – completely untreated cells (as the legend states) or SDF-1+ cells? The bars that are showing significance do not differ from the untreated control that is black bar. Please explain.
R: The negative control consists in untreated cells not chemo-attracted by SDF-1 (black bar). The non-treated cells chemo-attracted by SDF-1 (the dark gray bars) represent our positive control, since we investigated the effect of CD73 inhibitors on migration induced by SDF-1, we indicated only the significance between treated cells vs the positive control. However, as suggested, we also added the significance between the negative control and SDF-1 stimulated cells.
Reviewer 2 Report
Summary
The authors present an interesting and well written report addressing the somewhat confusing and unresolved signaling by adenosine in lung cancers and associated with stromal cells. The research utilizes lung cancer sphere models to understand how adenosine signaling antagonists can specifically promote lung cancer progression. The authors go on to show how adenosine from the bone stromal cells results in a unique tumor phenotype. The authors also go on to show that adenosine affects osteoblasts and osteoclasts uniquely which aid in stabling the bone pre-metastatic niche which is important site of metastasis for lung cancers. Prior to publication I would suggest some significant considerations for the authors to enhance the information conveyed to their targeted audience. The greatest concern I have is how appropriate in vivo or clinical samples databases can correlate these studies and should be discussed in greater detail.
Specific concerns:
- Appropriate statistical tests and comparisons should be made and justified. It’s not always clear what is being directly compared by a paired or ANOVA analysis.
- Is there are better way to characterize the growth rates of the spheres in size and number? Can the authors comment on whether the seeding densities affects the rate of cell aggregation or whether a single cell is sufficient to generate the CSC spheres?
- Rigor should be stated for how many times any given experiment was repeated or replicated.
Author Response
We thank the Editor and the Reviewers for handling and carefully revising our submission and for providing valuable suggestions that helped us to improve the manuscript.
In the revised version we have tried to address all points and concerns raised by the reviewers, by better elucidating and explaining technical issues and data interpretation.
Please find below a point-by-point reply to reviewers’ questions.
Summary
The authors present an interesting and well written report addressing the somewhat confusing and unresolved signaling by adenosine in lung cancers and associated with stromal cells. The research utilizes lung cancer sphere models to understand how adenosine signaling antagonists can specifically promote lung cancer progression. The authors go on to show how adenosine from the bone stromal cells results in a unique tumor phenotype. The authors also go on to show that adenosine affects osteoblasts and osteoclasts uniquely which aid in stabling the bone pre-metastatic niche which is important site of metastasis for lung cancers. Prior to publication I would suggest some significant considerations for the authors to enhance the information conveyed to their targeted audience. The greatest concern I have is how appropriate in vivo or clinical samples databases can correlate these studies and should be discussed in greater detail.
R: We specified at the end of the discussion the relevance of a potential association between checkpoint and Ado inhibitors, lines 414-421.
Specific concerns:
- Appropriate statistical tests and comparisons should be made and justified. It’s not always clear what is being directly compared by a paired or ANOVA analysis.
R: As requested, we have now better specified the statistical tests used for each comparison in the Figure Legends
- Is there are better way to characterize the growth rates of the spheres in size and number? Can the authors comment on whether the seeding densities affects the rate of cell aggregation or whether a single cell is sufficient to generate the CSC spheres?
R: Spheres were generated by seeding single cells at low density (10.000/ml), to avoid cellular aggregation and to assure the clonal origin of spheres. We previously demonstrated the clonal origin of spheres obtained in this culture condition in Bertolini et al PNAS 2009. We better specified this relevant point in Material and Methods “ NSCLC cells and CSC sphere cultures”
Rigor should be stated for how many times any given experiment was repeated or replicated.
R: Coherently to the request, we modified this aspect